# Smartphone Photogrammetric Assessment for Head Measurements

**DOI:** 10.3390/s23219008

**Published:** 2023-11-06

**Authors:** Omar C. Quispe-Enriquez, Juan José Valero-Lanzuela, José Luis Lerma

**Affiliations:** Photogrammetry and Laser Scanner Research Group (GIFLE), Department of Cartographic Engineering, Geodesy and Photogrammetry, Universitat Politècnica de València, Camino de Vera s/n, 46022 Valencia, Spain; juavalan@topo.upv.es (J.J.V.-L.); jllerma@cgf.upv.es (J.L.L.)

**Keywords:** 3D measurement, metric assessment, cranial deformation, smartphone device, 3D scanning

## Abstract

The assessment of cranial deformation is relevant in the field of medicine dealing with infants, especially in paediatric neurosurgery and paediatrics. To address this demand, the smartphone-based solution PhotoMeDAS has been developed, harnessing mobile devices to create three-dimensional (3D) models of infants’ heads and, from them, automatic cranial deformation reports. Therefore, it is crucial to examine the accuracy achievable with different mobile devices under similar conditions so prospective users can consider this aspect when using the smartphone-based solution. This study compares the linear accuracy obtained from three smartphone models (Samsung Galaxy S22 Ultra, S22, and S22+). Twelve measurements are taken with each mobile device using a coded cap on a head mannequin. For processing, three different bundle adjustment implementations are tested with and without self-calibration. After photogrammetric processing, the 3D coordinates are obtained. A comparison is made among spatially distributed distances across the head with PhotoMeDAS vs. ground truth established with a Creaform ACADEMIA 50 while-light 3D scanner. With a homogeneous scale factor for all the smartphones, the results showed that the average accuracy for the S22 smartphone is −1.15 ± 0.53 mm, for the S22+, 0.95 ± 0.40 mm, and for the S22 Ultra, −1.8 ± 0.45 mm. Worth noticing is that a substantial improvement is achieved regardless of whether the scale factor is introduced per device.

## 1. Introduction

The accurate assessment of morphological cranial deformation is important in the medical field [1], especially for experts such as paediatricians and paediatric neurosurgeons, for whom this information is relevant to studying the characteristics of plagiocephaly [2,3,4]. Some authors have discovered evidence of an association between plagiocephaly and developmental delay [5], making precise detection in the early stages crucial, as early intervention may assist in addressing the condition [6,7].

In recent decades, the accuracy and precision of 3D scanners have improved a lot. The 3D scanner devices have their own set of advantages and limitations, not only in terms of accuracy but also in terms of data acquisition, processing time, possible applications, and user friendliness [8]. Three-dimensional surface scanning devices represent a promising method for obtaining measurements for anthropometric cranial shape [9]. Three-dimensional scanners have gained widespread usage for accurately creating 3D body models [10,11,12]. Nevertheless, these high-end devices often come with a high cost and necessitate a controlled environment for optimal operation.

In recent times, there has been a growing interest in the utilisation of mobile devices such as smartphones and tablets for data capture and the assessment of cranial deformities. In this context, PhotoMeDAS [13] is introduced, which relies on the primary camera of the device and sticker recognition on a cap. This mobile application was specifically developed for the Android operating system, despite the fact that iOS implementation is being carried out at present.

Another option for diagnosis is the integration of artificial intelligence for determining plagiocephaly in mobile applications [14,15]. Nevertheless, further research, namely extensive, high-quality training datasets, are still pending to achieve high statistical estimates. Regarding data acquisition, other alternatives for measuring cranial deformities with active sensors (TrueDepth technology integrated into high-end iOS mobile devices) have also been explored [16].

Photogrammetric Medical Deformation Assessment Solutions (PhotoMeDAS, https://PhotoMeDAS.eu, accessed on 9 November 2022, Universitat Politècnica de València, València, Spain) was developed as a cost-effective tool for measuring and evaluating cranial deformation in infants [17]. This innovative solution consists of a coded cap that is placed on the infant’s head, a smartphone application used to record the head cap’s position relative to its barycenter, and a cloud-based processing system that automatically generates the 3D model of the head along with anthropometric deformational indices.

The objective of this study is to assess the effectiveness and accuracy of the 3D measurements obtained with PhotoMeDAS using various smartphones. In fact, one might expect no variations in performance among different smartphones. But this fact needs to be demonstrated. The accuracy of the derived measurements in three mobiles and with three different bundle adjustment approaches will be examined and compared against a reference 3D model acquired through the ACADEMIA 50 3D scanner (manufactured by Creaform in Levis, QC, Canada), known for its accurate head modelling capabilities [10,11]. Furthermore, the consistency among different mobile devices will be evaluated. This analysis will be crucial in validating the clinical utility of the 3D models generated with PhotoMeDAS on different mobile devices.

## 2. Materials and Methods

### 2.1. PhotoMeDAS App

The low-cost photogrammetric solution for smartphones, PhotoMeDAS (version 1.7), requires a coded cap (Figure 1a) to extract the 3D points that will be used for mesh creation. For this investigation, the cap was placed on a mannequin head (Figure 1b,c). This particular head is a bald mannequin used in the professional cosmetology industry. The model gender is female and bears the brand name “OLD STREET”. Its height is 24 cm, with a head circumference of 50 cm and a weight of 380 g.

The PhotoMeDAS app (Figure 1d) features an interface similar to recording a video. To perform cranial measurements, the smartphone is pointed at the patient’s entire head, moving it around appropriately, following along or across strips. The application detects the visible targets of the coded cap in each frame and records their positions. Once the entire head has been covered, the app notifies the user and uploads the data for autonomous processing.

PhotoMeDAS uses a coordinate system where the *y*-axis is defined by both preauricular points, the *x*-axis is defined by half of the preauricular points and the frontal point, and the *z*-axis is such that a right-hand coordinate system is formed [13].

### 2.2. Smartphones

For this research on the metric performance of the PhotoMeDAS app, three smartphones were selected that meet identical state-of-the-art hardware and software available in the market: the Samsung Galaxy S22 Ultra, the Samsung Galaxy S22+, and the Samsung Galaxy S22. These devices have undergone an evaluation to ensure their suitability and ability to run the PhotoMeDAS app smoothly. Table 1 present a visual representation of the smartphone features related to the investigation for each of the selected mobile devices.

The Android-compatible version of PhotoMeDAS has been launched with the aim of enhancing its accessibility and relevance in scientific research and applications related to clinical data acquisition and paediatric medical diagnosis during consultation. The state-of-the-art Samsung Galaxy S22, S22+, and S22 Ultra smartphone models have been chosen due to their prominent position in the Android phone market and their diverse range of technical features such as dimensions, screen size, screen resolution, and weight from a single manufacturer (vid. Table 1), despite the fact that the three models keep the same roots, i.e., processor, imaging sensor, and ultra-wide-angle lens for the rear camera. In addition, the choice was also rooted in the exploration of alternatives to the iPhone’s smartphones [14,16].

### 2.3. 3D Scanner

The Creaform ACADEMIA 50 3D scanner was used to obtain an accurate 3D model of the mannequin head. As a standard validated instrument, it can be considered the object’s ground truth. The ACADEMIA 50 3D scanner (Figure 2), projects a random light pattern onto the object. It is easy to set up and use. In principle, it is possible to scan objects made of any material, colour, or type of surface. Its technical specifications highlight its performance levels, with an accuracy of up to 0.250 mm and a measurement resolution of up to 0.250 mm [13].

The 3D scanner provided by the manufacturer included a calibration certificate. In this process, a silver sphere with 5785 points was used. The results showed an average deviation of 0.021/−0.017 mm, with a standard deviation of 0.0310 mm and maximum and minimum values of 0.3616 mm and −0.5873 mm, respectively. Additionally, as an integral part of our data acquisition procedure, the equipment’s calibration was performed using the calibration board (Figure 2b) before starting any data acquisition. According to the manufacturer, this calibration includes a verification of the proper functioning of both the scanner and the software used. To ensure the consistency of the 3D scanning measurements, scan repeatability tests were also conducted, with the results detailed in Table 2.

### 2.4. Workflow

The workflow (Figure 3) was divided into three stages. The first stage tackles data acquisition of the mannequin head (Figure 2c) either with smartphones running the PhotoMeDAS app or the ACADEMIA 50 3D scanner (Figure 2a) that requires the calibration plate (Figure 2b). The second stage is related to 3D processing, in which three different bundle adjustment approaches are applied with or without camera self-calibration; in addition, the 3D scanning processing is presented. The third and last stage deals with spatial analysis: statistical evaluations are made considering the 3D point clouds obtained with the different procedures.

Each smartphone device (Table 1) is used for approximately 45 min, recording in automatic mode a total of 12 times the mannequin head with each smartphone. During the data acquisition process, the operator activated the PhotoMeDAS app and captured the data by moving concentrically around the head along two strips and a third crossing strip. In this article, the ArUco targets will be referred to as targets for abbreviation purposes. Subsequently, the data were uploaded to the website https://PhotoMeDAS.eu/ (Reviewed on October 11, 2023). for processing. From each PhotoMeDAS recording, a 3D point cloud file is obtained and used to generate both a 3D model and a cranial anthropometric report [13].

The same mannequin head is scanned after calibrating the ACADEMIA 50 3D scanner (Figure 2a) using the calibration plate (Figure 2b) provided by the supplier.

Three different processing approaches are considered based on the variability of the records for determining the 3D point cloud coordinates. In all cases, the photogrammetric open-source Apero-MicMac (version 1.0) software was run [18].

### 2.5. Processing

#### 2.5.1. Processing I

For each of the 12 recordings per smartphone, on-the-job calibration is carried out through relative orientation, considering a model with 8 degrees of freedom: 1 for principal distance, 2 for principal point, 2 for distortion centre, and 3 for radial polynomial coefficients. The results of this processing are the interior orientation parameters, the exterior orientation parameters, and the 3D point cloud coordinates (Figure 4).

#### 2.5.2. Processing II

First, an external camera calibration with 12 ideal convergent full-of-texture images with the same set of 8 degrees of freedom will be carried out. Once the camera calibration parameters are determined and fixed for the following stages, the relative orientation will be used to determine the exterior orientation parameters and the 3D point cloud coordinates (Figure 4).

#### 2.5.3. Processing III

An additional self-calibration stage is included in Processing II, trying to improve the results due to the data recording in autofocus mode. Now, the 8 degrees of freedom for the geometric camera calibration are not per camera but per image, i.e., the bundle adjustment considers local instead of global additional parameters. The results of this processing are the interior orientation parameters per image, the exterior orientation parameters, and the 3D point cloud coordinates (Figure 4).

### 2.6. 3D Scanning

#### 2.6.1. Data Capture with the 3D Scanner

The ACADEMIA 50 3D scanner was used to model the mannequin head. The blue circles indicated in Figure 5b are associated with the positioning system used for data acquisition, which relies on the object positioning, geometry, and texture of the model. We conducted repeated measurements on six occasions to assess the precision of the current device. All the processes were run in the VXelements software version 8.0.0 (Figure 5); the textured model was subsequently exported.

#### 2.6.2. Three-Dimensional Scanning Handling

For the photogrammetric assessment, the 3D point cloud coordinates are needed. Therefore, the 3D scanning model (Figure 5b) was imported into the Agisoft Metashape software v. 1.7. First, the ‘connected component size’ filter tool automatically identified and eliminated small groups of points that were not significantly related to the primary 3D model. This filter effectively removes noise in the 3D scan, as unwanted 3D points tend to form small groups. Then, in the second step, manual cleaning was performed by searching for and removing isolated points that did not belong to the primary 3D model, ensuring that the data were clean and ready for further evaluation. Later, each corner’s, x, y, and z coordinates were manually identified and measured in the 3D textured model (Figure 6).

#### 2.6.3. Calculation of 3D Distances

Four distances were measured from each recording once the closest targets to key cranial points had been identified to determine both the longest and shortest width and length mannequin distances: preauricular distance (Figure 7a); maximum width (Figure 7b); and two maximum length frontal-occipital distances (Figure 7c,d). Each target’s centre coordinates were determined by computing the average of the four corresponding corners. The 3D distance between targets was then computed using the Euclidean distance formula.

## 3. Results

The ACADEMIA 50 mean and standard deviation distances corresponding to the four mannequin head distances are presented in Table 2. The mean values will be considered the ground-truth distances to assess the estimates obtained with the three photogrammetric processing approaches presented in Section 2.3.

The results achieved after recording the mannequin head 12 times with the PhotoMeDAS app and running Processing I, Processing II, and Processing III are presented in Figure 8 and Appendix A for the smartphone Samsung Galaxy S22; in Figure 9 and Appendix A for the smartphone Samsung Galaxy S22+; and in Figure 10 and Appendix A for the smartphone Samsung Galaxy S22 Ultra.

### 3.1. Comparative Analysis

For the descriptive analysis, the information presented in Figure 8, Figure 9 and Figure 10 is used, which depicts the preauricular, lateral, maximum right, and maximum left distances. Table 3 displays the main statistical characteristics grouped by the type of photogrammetric process employed and the mobile device type.

For a better understanding of the variation between the PhotoMeDAS results and the reference distances obtained with the 3D scanner, Figure 11 is presented, which graphs the data indicated in Table 3.

### 3.2. Precision Calculation

To calculate the precision of the distances, standard indicators such as the Coefficient of Variation (CV) and the Relative Standard Deviation (RSD) are used. Both terms are essentially equivalent and provide a uniform and standardized way of measuring data dispersion. The CV and the RSD allow users to express relative variability in percentage terms, which is especially useful when comparing data sets with different scales or magnitudes. These indicators help us understand how dispersed or clustered the values are in relation to the mean.

The CV is calculated by the relationship between the Mean and the Standard Deviation. In other words, the CV is obtained by dividing the Standard Deviation by the Mean. The Relative Standard Deviation (RSD) is determined by multiplying the Coefficient of Variation (CV) by 100. In general, the lower the RSD, the lower the dispersion of the results, and the higher the precision. In Table 4, the calculations performed to obtain the RSD for each processing are summarised, categorised by the type of smartphone.

### 3.3. Accuracy Calculation

Accuracy in measurement is defined as the discrepancy between the obtained result and the reference value. To assess accuracy, we can express it in absolute or relative terms (percentage). To calculate the relative difference, also known as relative error, we apply the following formula:(1)Absolute Difference (AD)=|Measured Value−Reference Value|
(2)Relative Difference (RD)=|(Measured Value−Reference Value)/Reference Value|×100

In this equation, “Measured Value” represents the result obtained from the measurement, while “Reference Value” is the known reference or certified value, i.e., obtained herein with the ACADEMIA 50 3D scanner.

The relative difference provides a quantitative measure of the accuracy of our measurements. A low value of relative difference indicates high accuracy, meaning that the measured result is very close to the expected value. On the other hand, a high value of relative difference indicates lower accuracy, implying that the measured result significantly deviates from the expected value. In Table 5, the calculations performed to obtain the RD for each processing are summarised, categorised by the type of smartphone.

### 3.4. Relationship between Scaling Factors for S22, S22+, and S22 Ultra

The calculation of the scale factor was performed by relating the distances obtained with PhotoMeDAS to the reference distances obtained with the 3D scanner. This was conducted for each procedure and for the three smartphone models. Table 6 summarizes these scale factors.

According to the data from Table 6, it can be observed that the average scale factor correction for the mobile devices is as follows: for S22, it is 0.993; for S22+, 1.006; and for S22 Ultra, 0.988. These values indicate that the 3D coordinates generated by PhotoMeDAS tend to be slightly oversized in the case of S22 and S22 Ultra and slightly undersized for S22+.

To assess the impact of the average scale factors (Table 6) on the four distances obtained with PhotoMeDAS, each of the four distances (preauricular, lateral, maximum right, and maximum left) was multiplied by the corresponding scale factor. This process resulted in new distances, which were then compared with the reference values obtained from the 3D scanner, as presented in Table 7.

For a better understanding of the overall distance differences between the PhotoMeDAS results and the reference distances obtained with the 3D scanner, Figure 12 is presented, which graphs the data indicated in Table 7.

### 3.5. t-Student Test

A comparative evaluation of the *t*-Student test is carried out between the distances obtained directly with PhotoMeDAS without scale factor correction and the distances after scale factor correction. The IBM-SPSS Statistics (version 29.0) software was used, and the Shapiro Wilk normality test was performed for all cases, as well as the relationship of the “one-sample-test”. In this instance, a significance level of alpha 0.05 was utilised, and the two-tailed *p*-value was considered, as summarised in Table 8.

In the analysis comparing the four linear measurements obtained with both the 3D reference model and PhotoMeDAS, the ideal scenario is a zero difference. However, when the scale factor is not applied, the results yield significantly low *p*-values < 0.001, indicating that there is no statistically significant equivalence between the datasets. Conversely, after incorporating the scale factor, a reduction in the variation between both datasets are observed, approaching a value close to zero. This finding suggests a higher level of agreement and similarity between the results obtained with PhotoMeDAS and the 3D scanner; it is verified with a *p*-value > 0.05 (Table 8).

## 4. Discussion

Obtaining head measurements to assess deformities can be conducted using various instruments or methodologies, such as callipers and measuring tape, as well as using photogrammetric techniques [19,20]. Three-dimensional scanning technology is also used as a faster but more expensive solution [21,22,23,24], or even through a comprehensive solution, which is the use of mobile applications like PhotoMeDAS [13]. In the case of PhotoMeDAS, due to the involvement of various factors, it is pertinent to analyse their influence on the results. That is why in this section, we analyse the results, considering the type of processing, the various mobile devices employed, and the impact of incorporating the scale factor.

### 4.1. Evaluation of Smartphone Model and Photogrammetric Processing

The first evaluation is about the smartphone model. The analysis reveals up to a millimetre difference in distance measurements depending on the type of smartphone used and a variability of up to 0.6 mm. While the S22 shows undersized distances ($\bar{x}$: −1.1 mm and **σ**: 0.5 mm), the S22 Ultra presents even smaller distances ($\bar{x}$: −1.8 mm and **σ**: 0.4 mm), and the S22+ shows oversized distances ($\bar{x}$: 1 mm and **σ**: 0.6 mm).

In the second evaluation, we evaluate the impact of choosing the photogrammetric processing method (Processing I, II, and III). We compared the results presented in Table 3 and Figure 3. Based on these findings, it can be stated that the average standard deviation for Processing I and II is 0.5 mm, while for Processing III, it is 0.6 mm. In summary, we can conclude that the choice of photogrammetric processing has minimal impact on the variability of the measurements obtained.

For medical practitioners, it is important to make measurements that are considered close to 2 mm, as this measurement is not perceptible to the naked eye [25]. Therefore, the data provided by PhotoMeDAS is sufficient to evaluate metric results in baby heads.

### 4.2. Evaluation of Precision and Accuracy

Precision analysis of distance measurements made with various smartphone models and photogrammetry procedures shows RSD (Relative Standard Deviation) values below 0.5%. This result strongly supports the feasibility of using 3D photogrammetry with smartphones in applications related to the evaluation of cranial deformations (Table 4).

The accuracy analysis shows that the S22 has an average AD of 1.1 mm relative to reference values, with a RD of 0.7%. In the case of the S22+, the average AD is 1 mm, with a RD of 0.6%. However, the S22 Ultra features the highest average AD, with 1.8 mm and 1.2% RD. In summary, these results indicate that distance measurements made by 3D photogrammetry with smartphones are highly reliable and can be used for the evaluation of cranial deformations, with minimal relative differences compared to the values obtained by a reference 3D scanner (Table 5).

In [13], which used the 2022 version of PhotoMeDAS, it was found that the 3D models generated had uncertainties in the coordinates of up to 1.5 mm. In the publication [7], it was observed that the variance in the measurements was minimal, with a maximum fluctuation of approximately 2 mm both between observers (interobserver) and within the same observer at different times (intraobserver). In contrast, according to the results of the experimentation presented in Table 5, there is an overall average of 1.3 mm for AD (S22, S22+, and S22 Ultra). While this analysis considers the average for comparison with [7,13], it is pertinent to mention the existence of a range of minimum and maximum distance values for a more detailed future evaluation. For instance, a distance variation ranging from −2.5 mm to −0.1 mm was observed, as exemplified by the case of the S22 when using Processing I (Table 3).

The fact that the three high-end smartphone models provide different accuracies (Figure 11) for the same head model generates an important observation that highlights the variability in measurements among mobile devices. This underscores the importance of scaling as an effective strategy to standardise results and measurements (Figure 12), ensuring consistency and data reliability regardless of the smartphone model used. More and more, smartphones are built by different manufacturers using a variety of specifications for hardware (imaging sensors, lenses…) and software components.

### 4.3. Evaluation of Scaling Factor

Prior to applying the scaling factor, we present the results in Table 6. The average scaling factor obtained for the S22 smartphone was 0.993, and for the S22 Ultra, it was 0.988. In both cases, these scaling factors are less than 1. In contrast, for the S22+ smartphone, the scaling factor was 1.006, which is greater than 1. This latter scaling factor is related to the result presented in [13], exceeding on average 1.01 times the corresponding ground truth.

The scaling factors are generally close to 1, but they can be greater or smaller depending on the type of smartphone, which influences the resulting measurements.

To assess the influence of the scaling factor, we have presented the results in Table 7. When the average scaling factor is applied, based on the smartphone model, the differences between the distances obtained with the scanner and PhotoMeDAS tend to approach zero, as illustrated in Figure 12. This contributes to making the results consistent and the measurements not excessively oversized or undersized.

Therefore, it is strongly recommended that in future research, the scaling factor be determined and applied according to the smartphone model used. This helps reduce uncertainty related to the possibility of measurements being oversized or undersized.

The Student’s *t*-test, as presented in Table 8, reveals that the application of scale factors has a statistically significant impact on the accuracy of measurements. *p*-values approaching zero indicate a substantial improvement in measurement consistency when scale factors are utilised. This underscores the importance of incorporating scale factors into photogrammetry to correct for systematic discrepancies and align measurements with a common reference.

Finally, to assess the improvement from the individualised scale factor application, it is noted that an average is computed with absolute values. Without the scale factor, the result is $\bar{x}$: 1.3 mm and σ: 0.5 mm (Table 3), whereas with the application of the scale factor, the result is $\bar{x}$: 0 mm and σ: 0.5 mm (Table 7).

### 4.4. Limitations and Future Areas of Research

A limitation of our study is the relatively small sample size of the mobile devices tested; three present high-end smartphones from the same manufacturer. While we looked at three smartphone models, a wider range of devices from different manufacturers could provide a broader understanding of the potential variability introduced by different smartphones.

In addition, results are shown on a single symmetric synthetic head, and cranial asymmetry is not evaluated in this study. Future research should aim to include a more diverse set of samples, including cases of infants under the age of 3, despite the fact that it is rarely admissible to measure 12 times any infant by different smartphone models.

An aspect to consider in future developments is the integration of artificial intelligence [15] into the processing phase. For future research, the data capture system could be optimised using feedback mechanisms [16] to ensure the correct distance between the head and the smartphone. This would involve utilising sound, vibration, or visual cues on the smartphone’s screen to streamline and expedite the process.

Finally, our study identified differences in measurement accuracy between smartphone models but did not delve into the specific reasons for these discrepancies. Investigating the factors contributing to these variations could provide valuable information for future research.

## 5. Conclusions

In summary, our study on linear head assessment using photogrammetry and different smartphones yields important conclusions. The extensive analysis presented herein supports the PhotoMeDAS photogrammetric solution as a reliable development for extracting linear measurements comparable to state-of-the-art 3D scanning devices used in medical applications.

The importance of proper application of the scale factor is considered an essential decision to achieve zero-bias linear measurements. This point is essential to ensure that the actual dimensions of objects are properly represented in 3D models, contributing to reliable measurements of head parameters and, thus, cranial deformations.

This study has revealed that different smartphones, even from the same manufacturer, can introduce variations in the precision and accuracy of measurements. It is essential to take these differences into account when selecting mobile devices for clinical and research applications. Therefore, it is not recommended to change smartphones during a clinical consultation.

The advantage of using photogrammetry with smartphones for monitoring cranial deformations relies on the minimum resources required and the fast processing. This non-invasive and accessible approach may facilitate more frequent and earlier assessment of cranial deformities in infants, which in turn could improve treatment planning.

## Figures and Tables

**Figure 1 sensors-23-09008-f001:**
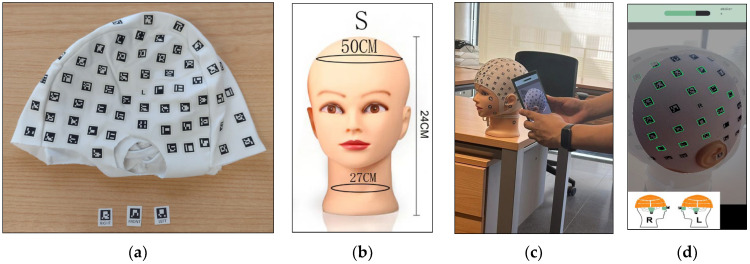
Materials: (**a**) PhotoMeDAS coded cap and three targets; (**b**) Mannequin head; (**c**) Data acquisition; (**d**) PhotoMeDAS app during data acquisition.

**Figure 2 sensors-23-09008-f002:**
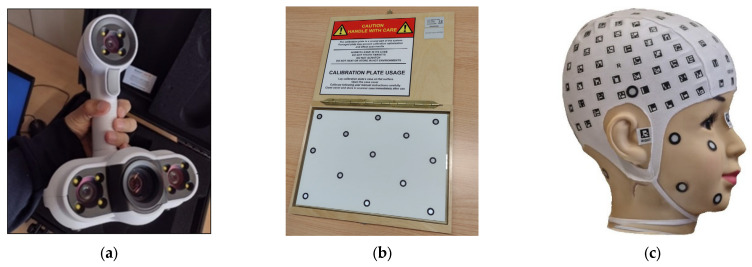
Scanning setup: (**a**) ACADEMIA 50; (**b**) Scanner calibration plate; (**c**) Coded cap with additional round retroreflective targets.

**Figure 3 sensors-23-09008-f003:**
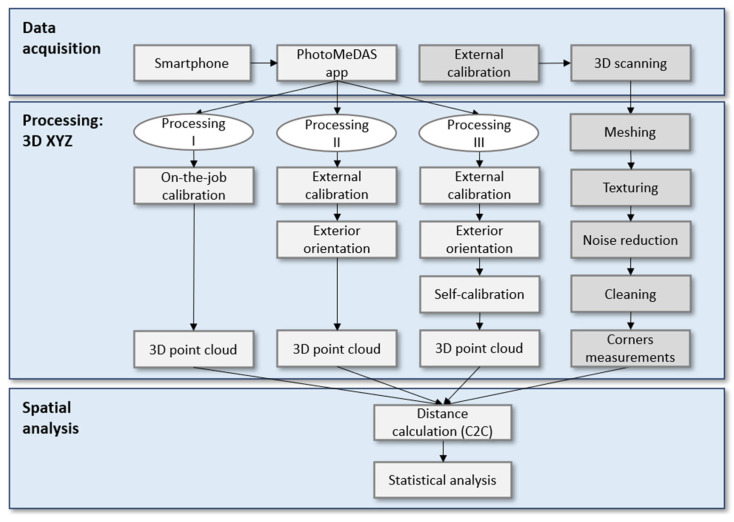
Workflow schema.

**Figure 4 sensors-23-09008-f004:**
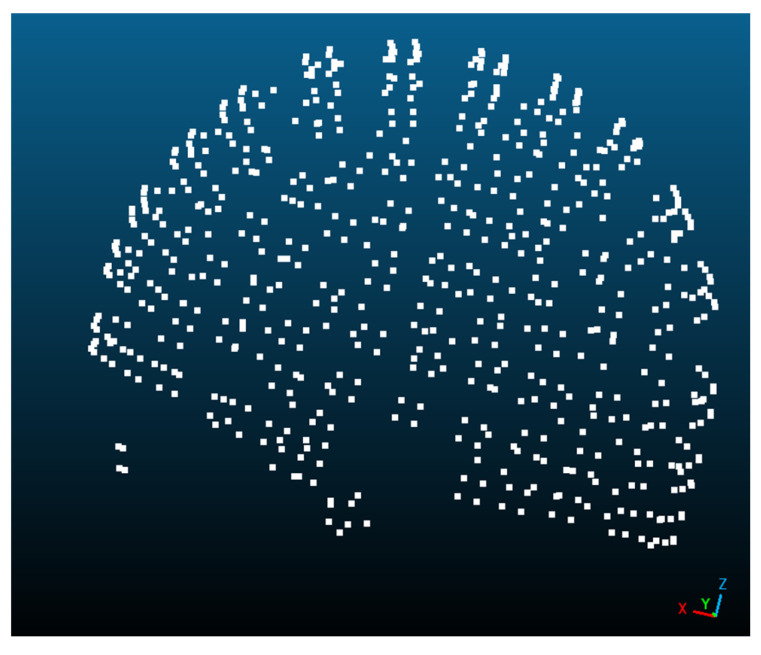
Three-dimensional point cloud visualisation in CloudCompare.

**Figure 5 sensors-23-09008-f005:**
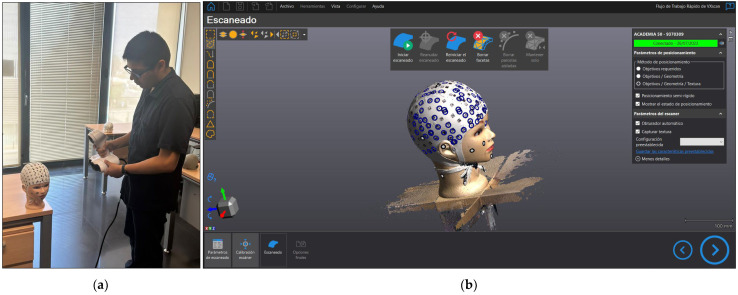
ACADEMIA 50: (**a**) Data acquisition; (**b**) Print-out of the data acquisition in VXelements software.

**Figure 6 sensors-23-09008-f006:**
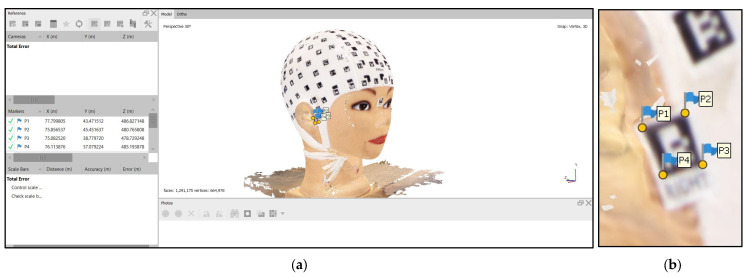
Virtual 3D model: (**a**) Three−dimensional scanning model imported into Agisoft Metashape; (**b**) Measurement of the corner target coordinates.

**Figure 7 sensors-23-09008-f007:**
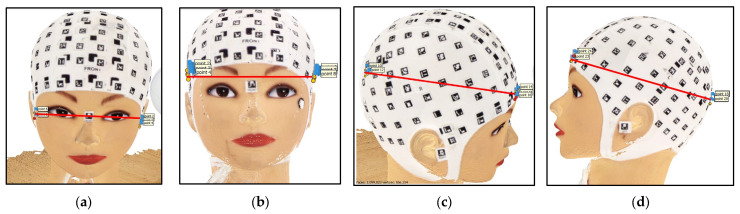
Measured distances: (**a**) preauricular distance; (**b**) lateral distance; (**c**) maximum length frontal-occipital right distance; (**d**) maximum length frontal-occipital left distance.

**Figure 8 sensors-23-09008-f008:**
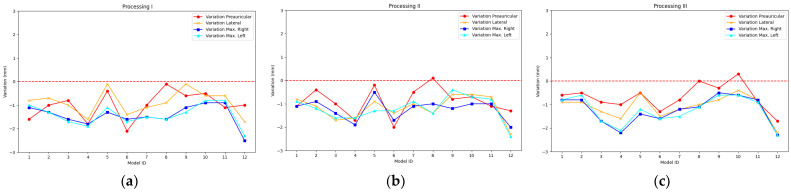
Accuracy bias for Galaxy S22: (**a**) Processing I, (**b**) Processing II, and (**c**) Processing III.

**Figure 9 sensors-23-09008-f009:**
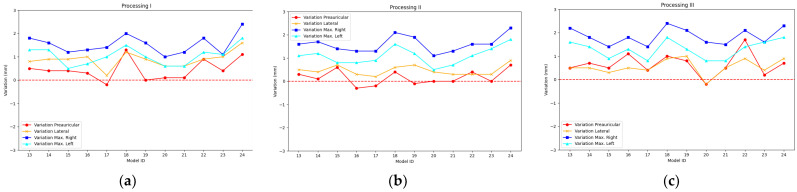
Accuracy bias for Galaxy S22+: (**a**) Processing I, (**b**) Processing II, and (**c**) Processing III.

**Figure 10 sensors-23-09008-f010:**
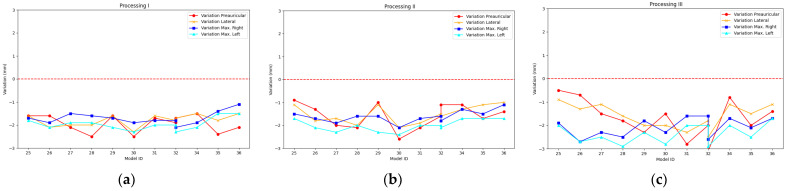
Accuracy bias for Galaxy S22 Ultra: (**a**) Processing I, (**b**) Processing II, and (**c**) Processing III.

**Figure 11 sensors-23-09008-f011:**
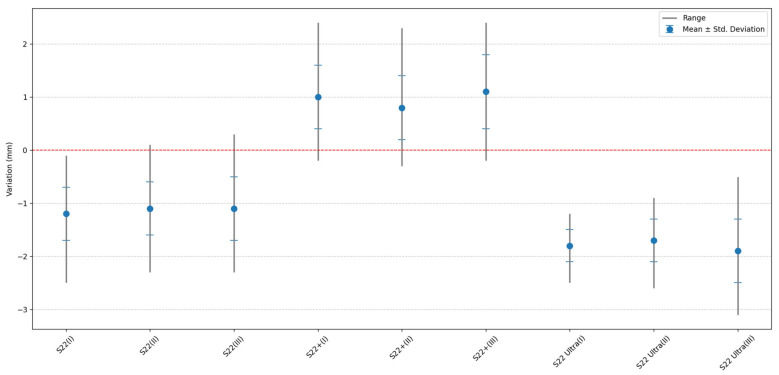
Media, Standard Deviation, Minimum, and Maximum by smartphone and procedure for S22, S22+, and S22 Ultra: (**I**) Processing I, (**II**) Processing II, and (**III**) Processing III.

**Figure 12 sensors-23-09008-f012:**
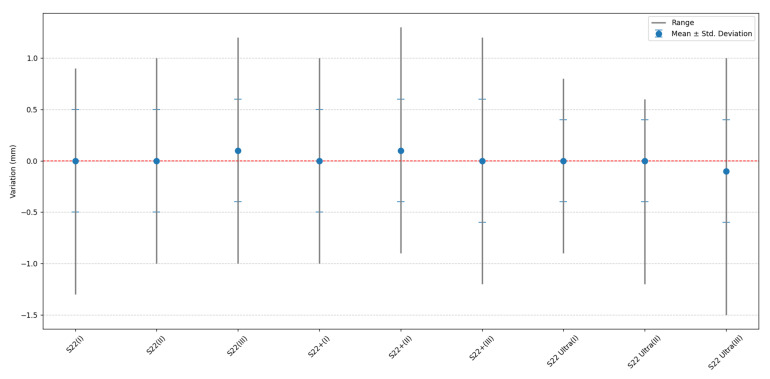
Distance differences after correcting the scale factor by model and procedure for S22, S22+, and S22 Ultra: (**I**) Processing I, (**II**) Processing II, and (**III**) Processing III.

**Table 1 sensors-23-09008-t001:** Comparison of Samsung Galaxy S22, S22+, and S22 Ultra Specifications (limited to the features related to this study).

Characteristic	S22	S22+	S22 Ultra
Processor	**CPU Speed**: 1.8 GHz; **CPU Type**: Octa-Core		
Rear camera—Ultra wide angle	12 MP F2.2 [FF], FOV 120°, 1/2.55”, 1.4 µm	12 MP F2.2 [FF], FOV 120°, 1/2.55”, 1.4 µm	12 MP F2.2 [Dual Pixel AF], FOV 120°, 1/2.55”, 1.4 µm
Rear camera—Wide Angle	50 MP F1.8 [Dual Pixel AF], OIS, FOV 85°, 1/1.56”, 1.0 µm with Adaptive Pixel	50 MP F1.8 [Dual Pixel AF], OIS, FOV 85°, 1/1.56”, 1.0 µm with Adaptive Pixel	108 MP F1.8 [PDAF], OIS, FOV 85°, 1/1.33”, 0.8 µm with Adaptive Pixel
Rear CAMERA—Telephoto lens	10 MP F2.4 [3x, PDAF], OIS FOV 36°, 1/3.94”, 1.0 µm	10 MP F2.4 [3x, PDAF], OIS FOV 36°, 1/3.94”, 1.0 µm	10 MP F2.4 [3x, Dual Pixel AF], OIS, FOV 36°, 1/3.52”, 1.12 µm
Physical specifications	**Dimentions**: 146.0 mm × 70.6 mm × 7.6 mm; **Weight**: 167 g	**Dimentions**: 157.4 mm × 75.8 mm × 7.6 mm; **Weight**: 195 g	**Dimentions**: 163.3 mm × 77.9 mm × 8.9 mm; **Weight**: 228 g
Screen	**Resolution**: 2340 × 1080 (FHD+); **Size**: 153.9 mm (6.1” full rectangle)/149.9 mm (5.9” rounded corners) **Technology: Dynamic** AMOLED 2X **Number of colours**: 16 M	**Resolution**: 2340 × 1080 (FHD+) **Size**: 166.5 mm (6.6” full rectangle)/162.1 mm (6.4” rounded corners) **Technology: Dynamic** AMOLED 2X **Number of colours**: 16 M	**Resolution**: 3088 × 1440 (Quad HD+); **Size**: 173.1 mm (6.8” full rectangle)/172.5 mm (6.8” rounded corners) **Technology: Dynamic** AMOLED 2X **Number of colours:** 16 M
Battery	**Internet usage time (4G):** Up to 15 h; **Battery capacity:** 3700 mAh	**Internet usage time (4G):** Up to 19 h; **Battery capacity:** 4500 mAh	**Internet usage time (4G):** Up to 19 h; **Battery capacity:** 5000 mAh

Source 1: https://www.samsung.com/es/smartphones/galaxy-s22-ultra/models/ (accessed on 11 October 2023). Source 2: https://www.samsung.com/es/support/mobile-devices/check-out-the-new-camera-functions-of-the-galaxy-s22-series/, (accessed on 11 October 2023).

**Table 2 sensors-23-09008-t002:** Summary of the sessions with the ACADEMIA 50 3D scanner after measuring the four Euclidean head distances.

Session	Preauricular Distance (mm)	Max. Length Distance Right (mm)	Max. Length Distance Left (mm)	Maximum Width Distance (mm)
Session 1	127.224	170.814	169.473	140.279
Session 2	127.176	170.845	169.510	140.226
Session 3	127.239	170.878	169.528	140.251
Session 4	127.165	170.845	169.504	140.111
Session 5	127.130	170.930	169.469	140.110
Session 6	127.193	170.669	169.491	140.127
**Minimum**	127.130	170.669	169.469	140.110
**Maximum**	127.239	170.930	169.528	140.279
**Mean**	**127.188**	**170.830**	**169.496**	**140.184**
**Standard deviation**	0.040	0.088	0.023	0.076

**Table 3 sensors-23-09008-t003:** Mean, standard deviation, maximum, and minimum distance for PhotoMeDAS are grouped by type of photogrammetric processing and smartphone model.

Smartphone	Process	Overall Results of Distance Differences			
		$\bar{x}$ (mm)	σ (mm)	Minimum (mm)	Maximum (mm)
**S22**	Processing I	−1.2	0.5	−2.5	−0.1
	Processing II	**−1.1**	**0.5**	−2.3	0.1
	Processing III	−1.1	0.6	−2.3	0.3
**S22+**	Processing I	1.0	0.6	−0.2	2.4
	Processing II	**0.8**	**0.6**	−0.3	2.3
	Processing III	1.1	0.7	−0.2	2.4
**S22 Ultra**	Processing I	**−1.8**	**0.3**	−2.5	−1.2
	Processing II	−1.7	0.4	−2.6	−0.9
	Processing III	−1.9	0.6	−3.1	−0.5

**Table 4 sensors-23-09008-t004:** RSD for the three photogrammetric processes and by type of smartphone.

Smartphone	Distance	Processing I			Processing II			Processing III		
		$\bar{x}$ (mm)	σ (mm)	RSD (%)	$\bar{x}$ (mm)	σ (mm)	RSD (%)	$\bar{x}$ (mm)	σ (mm)	RSD (%)
**S22**	**Preauricular**	128.2	0.6	0.5	128.1	0.6	0.5	127.9	0.6	0.4
	**Lateral**	141.1	0.5	0.4	141.4	0.4	0.3	141.3	0.5	0.4
	**Max right**	172.2	0.4	0.3	172.0	0.5	0.3	172.0	0.6	0.3
	**Max left**	170.9	0.4	0.3	170.7	0.5	0.3	170.7	0.6	0.3
	**Mean RSD**			**0.3**			**0.3**			0.4
**S22+**	**Preauricular**	126.8	0.5	0.4	127.0	0.3	0.2	126.5	0.5	0.4
	**Lateral**	139.3	0.3	0.2	139.7	0.2	0.2	139.7	0.3	0.2
	**Max right**	169.3	0.4	0.2	169.2	0.4	0.2	169.0	0.4	0.2
	**Max left**	168.5	0.4	0.2	168.4	0.4	0.2	168.2	0.4	0.2
	**Mean RSD**			0.3			**0.2**			0.3
**S22 Ultra**	**Preauricular**	129.1	0.4	0.3	128.8	0.6	0.4	128.9	0.8	0.6
	**Lateral**	142.0	0.2	0.2	141.7	0.4	0.3	141.8	0.5	0.3
	**Max right**	172.5	0.3	0.2	172.4	0.3	0.2	172.9	0.4	0.2
	**Max left**	171.5	0.3	0.2	171.5	0.3	0.2	171.8	0.4	0.2
	**Mean RSD**			**0.2**			0.3			0.4

**Table 5 sensors-23-09008-t005:** Accuracy results for the three photogrammetric processes and by type of smartphone.

Smartphone	Distance	Processing I			Processing II			Processing III		
		$\bar{x}$ (mm)	AD (mm)	RD (%)	$\bar{x}$ (mm)	AD (mm)	RD (%)	$\bar{x}$ (mm)	AD (mm)	RD (%)
**S22**	**Preauricular**	128.2	1.0	0.79	128.1	0.9	0.71	127.9	0.5	0.39
	**Lateral**	141.1	0.9	0.64	141.4	1.2	0.86	141.3	1.1	0.78
	**Max right**	172.2	1.4	0.82	172.0	1.2	0.70	172.0	1.2	0.70
	**Max left**	170.9	1.4	0.83	170.7	1.2	0.71	170.7	1.2	0.71
	**Mean**		1.2	0.77		**1.1**	0.74		**1.1**	**0.65**
**S22+**	**Preauricular**	126.8	0.4	0.31	127.0	0.2	0.16	126.5	0.9	0.71
	**Lateral**	139.3	0.9	0.64	139.7	0.5	0.36	139.7	0.5	0.36
	**max right**	169.3	1.5	0.88	169.2	1.6	0.94	169.0	1.8	1.05
	**max left**	168.5	1.0	0.59	168.4	1.1	0.65	168.2	1.3	0.77
	**Mean**		**0.9**	0.61		**0.9**	**0.52**		1.1	0.72
**S22 Ultra**	**Preauricular**	129.1	1.9	1.49	128.8	1.6	1.26	128.9	1.5	1.18
	**Lateral**	142.0	1.8	1.28	141.7	1.5	1.07	141.8	1.6	1.14
	**Max right**	172.5	1.7	1.00	172.4	1.6	0.94	172.9	2.1	1.23
	**Max left**	171.5	2.0	1.18	171.5	2	1.18	171.8	2.3	1.36
	**Mean**		1.9	1.24		**1.7**	**1.11**		1.9	1.23

**Table 6 sensors-23-09008-t006:** Overall average scale factors for the Samsung Galaxy S22, S22+, and S22 Ultra.

Smartphone	Processing	Variation Preauricular	Variation Lateral	Variation Max Right	Variation Max Left	Mean
**S22**	Processing I	0.992	0.994	0.992	0.992	**0.993**
	Processing II	0.993	0.992	0.993	0.993	**0.993**
	Processing III	0.995	0.992	0.993	0.993	**0.993**
**S22+**	Processing I	1.003	1.006	1.009	1.006	**1.006**
	Processing II	1.001	1.003	1.009	1.006	**1.005**
	Processing III	1.005	1.004	1.011	1.008	**1.007**
**S22 Ultra**	Processing I	0.985	0.988	0.99	0.989	**0.988**
	Processing II	0.988	0.989	0.991	0.988	**0.989**
	Processing III	0.987	0.989	0.988	0.986	**0.988**

**Table 7 sensors-23-09008-t007:** Mean, standard deviation, maximum, and minimum overall distance differences between the reference value and PhotoMeDAS distances after correcting the factor scale are grouped by type of photogrammetric processing and smartphone model.

Smartphone	Processing	Overall Results			
		$\bar{x}$ (mm)	σ (mm)	Minimum (mm)	Maximum (mm)
**S22**	Processing I	**0.0**	**0.5**	−1.3	0.9
	Processing II	**0.0**	**0.5**	−1.0	1.0
	Processing III	0.0	0.6	−1.2	1.2
**S22+**	Processing I	**0.0**	**0.5**	−1.0	1.0
	Processing II	0.1	0.5	−0.9	1.3
	Processing III	0.1	0.5	−1.0	1.2
**S22 Ultra**	Processing I	0.1	0.5	−1.0	1.2
	Processing II	**0.0**	**0.4**	−1.2	0.6
	Processing III	−0.1	0.5	−1.5	1.0

**Table 8 sensors-23-09008-t008:** *p*-values without and with scale factor correction.

Smartphone	Processing	*p* Value of 2 Queues	
		Without Scaling	After Scaling
**S22**	Processing I	<0.001	0.512
	Processing II	<0.001	0.896
	Processing III	<0.001	0.872
**S22+**	Processing I	<0.001	0.490
	Processing II	<0.001	0.548
	Processing III	<0.001	0.794
**S22 Ultra**	Processing I	<0.001	0.118
	Processing II	<0.001	0.830
	Processing III	<0.001	0.192

## Data Availability

The data presented in this study are available on request from the group leader, Prof. José Luis Lerma.

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
