# Peer review of "Smartphone Photogrammetric Assessment for Head Measurements"

_sensors, 2023, doi:10.3390/s23219008_

Round 1
Reviewer 1 Report
Comments and Suggestions for Authors
Summary:
This article evaluates the precision of various mobile phone 3D systems in reconstructing a plastic head, using a proprietary industrial system PhotoMeDAS which targets the application of mobile phone measurement of cranial deformations using a specialized target cap.
Here the accuracy of 3D reconstruction is evaluated against a ground-truth 3D scanner (ACADEMIA 50) for several Samsung phones. 1-2 mm precision is obtained.
The accuracy report is interesting.
Comments:
The work does not actually assess cranial deformation, simply accuracy evaluations.
Limitations should specify that:
* results are shown on a single symmetric synthetic head, cranial asymmetry is not evaluated.
Abstract:
Remove hyperbolic, unscientific language,
e.g. remove “revolutionary”
“a revolutionary smartphone-based solution”
=> “a smartphone-based solution”
“A comparison is made among spatially distributed distances across the head with PhotoMeDAS and a 3D scanner that will be used as ground truth data“
mention the ACADEMIA 50 3D scanner
=>
“A comparison is made among spatially distributed distances across the head with PhotoMeDAS vs ground truth established with a ACADEMIA 50 3D scanner.“
Literature review:
Should include and discuss other mobile phone-based cranial measurement technologies [a], e.g. without specialized targeting systems. Note that computing cranial asymmetry indices does not require absolution mm coordinates.
[a] Watt, Ayden, et al. "Smartphone Integration of Artificial Intelligence for Automated Plagiocephaly Diagnosis." Plastic and Reconstructive Surgery Global Open 11.5 (2023).
Comments on the Quality of English LanguageNo problem.
Reviewer 2 Report
Comments and Suggestions for Authors
The manuscript aims to examine the potential of using smartphones for cranial deformation assessments, which is a timely and relevant topic given the increasing demand for low-cost, portable solutions in medical imaging. The study compares the accuracy of three different smartphone models for 3D imaging against a standard 3D scanner, which serves as the ground truth. Overall, the paper is well-structured, and the methods are rigorously described.
Materials and Methods - Workflow: While the workflow is laid out clearly, more information on why particular smartphones were chosen for the study would be beneficial.
Materials and Methods - 3D Scanning: It is essential to clarify how the accuracy of the 3D scanner itself was verified. While it is used as ground truth, understanding its limitations is important for contextualizing the results.
Results - Accuracy: It might be useful to discuss how the variation in accuracy across different smartphones can impact real-world applications.
Real-World Applications and Discussion: It would be beneficial to discuss the feasibility of these methods in actual clinical settings. For example, could head movement be an issue when using this method on live subjects? Including this would address practical challenges that could be encountered.
Round 2
Reviewer 2 Report
Comments and Suggestions for Authors
The authors have addressed all my concerns.